# Pre-Exposure Prophylaxis Uptake, Implementation and Barriers in Africa: A Scoping Review Protocol

**DOI:** 10.3390/ijerph22081300

**Published:** 2025-08-20

**Authors:** Silingene Joyce Ngcobo, Tracy Zhandire

**Affiliations:** Discipline of Nursing and Public Health, University of KwaZulu-Natal, Durban 4001, South Africa; ngcobos5@ukzn.ac.za

**Keywords:** pre-exposure prophylaxis, PrEP, HIV prevention, Africa, uptake barriers, implementation, scoping review protocol

## Abstract

Background: Despite significant advancements in HIV prevention, Africa continues to bear a disproportionately high burden of new infections. Pre-exposure prophylaxis (PrEP) has demonstrated over 90% efficacy in preventing HIV acquisition when taken consistently; however, its implementation and uptake across African countries remain suboptimal. Objective: This scoping review aims to systematically map existing literature on PrEP uptake and implementation in Africa, identify key barriers and facilitators influencing access and adherence, examine targeted population groups, and explore policy and programmatic approaches to PrEP delivery across diverse African contexts. Methods: The review will follow the methodological framework proposed by Arksey and O’Malley, refined by Levac et al., and will include comprehensive searches of electronic databases, grey literature, and official reports. Data will be thematically synthesized to identify research trends, gaps, and contextual variations. Expected Outcomes: The findings will provide a comprehensive overview of the current landscape of PrEP implementation in Africa, highlighting research trends, contextual variations across countries, and gaps in service delivery and policy frameworks. This will inform future implementation strategies, guide evidence-based approaches to enhance PrEP uptake, and support policymaking to scale up effective interventions across diverse African settings, ultimately contributing to HIV prevention efforts on the continent.

## 1. Introduction

### 1.1. The Current Landscape of HIV Prevention and PrEP in Africa

Human Immunodeficiency virus (HIV) remains a significant public health challenge in Africa, accounting for approximately two-thirds of the global HIV burden [1]. Despite increased access to antiretroviral therapy (ART) and ongoing prevention efforts, new infections continue at concerning rates, with marked regional disparities across Eastern, Southern, Western, and Central Africa [2,3]. In response, the Joint United Nations Programme on HIV/AIDS (UNAIDS) has called for innovative, combination prevention strategies to meet the 2030 goal of ending AIDS as a public health threat [4].

Pre-exposure prophylaxis (PrEP), a biomedical intervention using antiretroviral medications in oral or injectable form to prevent HIV infection in HIV-negative individuals, has emerged as a highly effective tool. Studies report 90–99% efficacy with consistent use, particularly for oral PrEP [5]. The World Health Organization (WHO) endorsed PrEP for populations at substantial risk in 2015 and integrated it into prevention strategies by 2017 [6,7].

### 1.2. Implementation Challenges and Research Gaps

Pre-exposure prophylaxis implementation and uptake across African countries has varied widely, constrained by barriers such as stigma, limited awareness, provider hesitancy, and health system challenges [8,9,10,11]. Economic factors, including the cost of medications, laboratory monitoring, and service delivery, present additional barriers to widespread implementation, particularly in resource-constrained settings [12]. Conversely, facilitators like community-based interventions, family engagement, and decentralized service delivery have shown promise in supporting uptake, particularly among key populations such as adolescent girls and young women (AGYW) [11,13].

Despite growing interest, PrEP programming often remains uneven and concentrated on select groups, potentially neglecting other at-risk populations. The sustainability of PrEP programs, particularly those initially supported by external donors, remains a critical concern for long-term HIV prevention efforts across the continent [14]. A clearer understanding of the barriers, enablers, and policy contexts influencing PrEP delivery is critical to inform scalable, equitable, and context-specific interventions.

### 1.3. Purpose and Significance of the Scoping Review

This scoping review aims to systematically map existing literature on PrEP uptake and implementation across African countries. Specifically, it seeks to:Identify and map studies on PrEP uptake and implementation in Africa;Explore barriers and facilitators at individual, community, health system, and policy levels;Examine which population groups have been prioritized for PrEP interventions; andReview national and sub-national approaches to PrEP policy and health system integration.

By synthesizing current evidence, this review will inform the development of targeted, evidence-based strategies to enhance PrEP uptake, strengthen HIV prevention programming, and contribute to global HIV control goals. Understanding regional variations in implementation will help tailor interventions to specific contexts, while identifying successful models may facilitate knowledge transfer across different African settings, ultimately supporting more sustainable and effective PrEP delivery systems.

## 2. Materials and Methods

This scoping review will follow the six-stage methodological framework proposed by Arksey and O’Malley [15] and further refined by Levac et al. [16]. The review will be reported in accordance with the PRISMA-ScR (Preferred Reporting Items for Systematic reviews and Meta-Analyses extension for Scoping Reviews) guidelines [17]. The protocol is registered on Open Science Framework (OSF), osf.io/asy6g, accessed on 30 April 2025.

### 2.1. Stage 1: Identifying the Research Question

The primary research question guiding this review is: What is the scope of existing literature on PrEP uptake and implementation in Africa, including the identification of barriers and facilitators, targeted population groups, and policy and programmatic approaches to PrEP delivery? This question aims to map the extent, range, and nature of research activity related to PrEP in African contexts.

### 2.2. Stage 2: Identifying Relevant Studies

#### 2.2.1. Search Strategy

A comprehensive search strategy will be developed in collaboration with a research librarian to ensure systematic and exhaustive identification of relevant studies. Electronic databases to be searched include PubMed, Scopus, CINAHL, Web of Science, African Journals Online (AJOL), and Embase. In addition, grey literature sources will be systematically explored to capture unpublished and non-indexed studies, reports, and programmatic documents. The review will consider studies published from 2012 onwards, following the WHO is formal endorsement of PrEP as an HIV prevention strategy in September [18].

#### 2.2.2. Language Considerations

Studies published in English, French, Portuguese, and Arabic will be included to ensure comprehensive coverage across Anglophone, Francophone, Lusophone, and Arabic-speaking African regions. This multilingual approach acknowledges the linguistic diversity of Africa and aims to minimize language-based exclusion of relevant evidence. For non-English publications, team members with relevant language proficiency will conduct the initial screening and data extraction. When necessary, key sections of included non-English articles will be translated into English using professional translation services to ensure accurate interpretation and data synthesis. The review team will document any instances where language limitations may have affected the comprehensiveness of data extraction.

#### 2.2.3. Grey Literature Sources to Be Searched

African Index MedicusWorld Health Organization (WHO) publicationsMinistry of Health websites from African countriesInternational AIDS Society (IAS) conference abstractsConference on Retroviruses and Opportunistic Infections (CROI) abstractsPrEP implementation project reportsDissertations and theses databases (e.g., ProQuest Dissertations and Theses Global)UNAIDS reports and publicationsPEPFAR (U.S. President’s Emergency Plan for AIDS Relief) implementation reportsThe Global Fund to Fight AIDS, Tuberculosis and Malaria documents

#### 2.2.4. Search Terms

The search strategy will use a combination of controlled vocabulary (e.g., MeSH terms) and free-text terms. The keywords and phrases are presented in Table 1, though this list is not exhaustive.

### 2.3. Stage 3: Study Selection

This scoping review will use the Population, Concept, Context (PCC) framework Table 2, to guide the study selection process.

In addition to the PCC framework, the following inclusion and exclusion criteria will be applied to guide the study selection process for this scoping review:

#### 2.3.1. Inclusion Criteria

Studies will be included if they:Geographic Focus: Are conducted in African countries or settings.Population: Involve individuals eligible for or using PrEP, including but not limited to adolescent girls and young women, men who have sex with men, female sex workers, people who inject drugs.Concept: Address aspects of PrEP uptake, implementation, adherence, barriers, facilitators, or outcomes.Study Design: Utilize original research employing quantitative, qualitative, or mixed-methods approaches.Review Type: Include systematic reviews, scoping reviews, and narrative reviews specific to African contexts.Policy and Program Evaluations: Present policy analyses, program evaluations, or implementation reports relevant to PrEP delivery.Language: Are published in English, French, Arabic, or Portuguese, provided a translation is available.Publication Date: Were published from January 2012 to the present.

#### 2.3.2. Exclusion Criteria

Studies will be excluded if they:Focus exclusively on the clinical efficacy or pharmacological properties of PrEP, without addressing uptake, implementation, or contextual barriers/facilitators.Are conducted outside African countries or do not include African populations in multi-country studies.Are opinion pieces, editorials, or commentaries that lack empirical data or analytical depth.Are conference abstracts with insufficient methodological detail or results related to implementation aspects of PrEP.Focus exclusively on PrEP efficacy or clinical outcomes without addressing implementation factors.Are conducted outside the African continent.Are opinion pieces, editorials, or commentaries without substantial empirical or analytical content.Are conference abstracts with insufficient information to characterize implementation factors.

#### 2.3.3. Selection Process

Two reviewers will independently screen titles and abstracts for potential inclusion. Full texts of potentially eligible studies will then be retrieved and independently assessed against the inclusion criteria by the same two reviewers. Discrepancies will be resolved through discussion or consultation with a third reviewer if necessary. The selection process will be documented using the PRISMA 2020 flow diagram [19].

### 2.4. Stage 4: Data Charting

In accordance with the frameworks by Arksey and O’Malley [15], Levac, Colquhoun [16] and the PRISMA-ScR guidelines, we will develop a standardized data charting form, Table 3, to systematically extract and organize relevant information from the included studies. This form will be pilot-tested on a subset of studies to ensure consistency and comprehensiveness. Two reviewers will independently chart data from each study, with discrepancies resolved through discussion or consultation with a third reviewer if necessary.

#### 2.4.1. Additional Considerations

Before full-scale data extraction, pilot the charting form on a subset of studies to ensure clarity and comprehensiveness. This process can help identify any ambiguities or missing elements in the form.Conduct data extraction independently by two reviewers to minimize bias and errors. Discrepancies should be resolved through discussion or consultation with a third reviewer.Data charting is an iterative process. As new themes or categories emerge, the charting form should be updated accordingly to capture all relevant information.

#### 2.4.2. Data Management

Extracted data will be compiled into a centralized database using spreadsheet software (e.g., Microsoft Excel) and specialized tools like Rayyan to facilitate analysis. The data will be organized to enable thematic synthesis and the identification of patterns across studies.

##### Potential Bias and Mitigation Measures

Although scoping reviews do not typically assess risk of bias formally, potential biases such as publication, language, and selection bias may arise. To minimize these, we will use a comprehensive search strategy across multiple databases and grey literature, apply clear inclusion/exclusion criteria consistently, and conduct independent screening by two reviewers with conflicts resolved by consensus. The study selection process will be fully documented to enhance transparency.

### 2.5. Stage 5: Collating, Summarizing, and Reporting the Results

This stage involves synthesizing the extracted data to address the research questions, identify patterns, and highlight gaps in the literature. The following approaches will be employed:

#### 2.5.1. Descriptive Quantitative Analysis

Numerical summaries will be generated to provide an overview of study characteristics, including:Study design (e.g., randomized controlled trials and observational studies)Country and settingSample size and demographicsPrEP uptake and adherence ratesImplementation characteristics (e.g., delivery models and duration)

These summaries will be presented in tables or charts to facilitate comparison across studies.

#### 2.5.2. Qualitative Thematic Analysis

Thematic analysis will be conducted to synthesize qualitative findings on:Barriers to PrEP uptake and adherenceFacilitators of successful implementationImplementation strategies employed

Data will be coded, patterns and themes identified, and a conceptual framework will be developed to characterize PrEP implementation in African contexts.

#### 2.5.3. Geographic Mapping

A geographic map will be created to visualize the distribution of studies across African countries and regions. This will highlight areas with extensive research and identify regions with limited or no studies, informing future research priorities.

#### 2.5.4. Population-Specific Analysis

Implementation approaches, barriers, and facilitators will be compared across different target populations (e.g., adolescent girls and young women, men who have sex with men, sex workers, and sero-discordant couples). This analysis will reveal population-specific challenges and effective strategies.

#### 2.5.5. Temporal Analysis

Changes in PrEP implementation approaches and outcomes over time will be examined, particularly in relation to evolving global guidelines and evidence. This will provide insights into trends and the impact of policy changes on PrEP uptake and adherence.

#### 2.5.6. Presentation of Results

The findings will be presented using narrative summaries, tables, and charts. These presentations will address each review question and highlight key themes, patterns, and gaps in the literature.

#### 2.5.7. Identification of Gaps and Implications

The review will identify gaps in the literature, such as under-researched populations or regions, and discuss implications for future research, practice, and policy. Recommendations will be made to guide future studies and inform the development of PrEP programs in African contexts.

### 2.6. Stage 6: Consultation

Incorporating stakeholder perspectives is essential to enhance the relevance and applicability of scoping reviews. Following the guidance of the JBI Scoping Review Methodology Group, this review will adopt a co-creation approach, engaging key stakeholders throughout the process.

#### 2.6.1. Stakeholder Identification and Recruitment

Stakeholders will be purposefully selected to represent a diverse range of perspectives, including:Policy makers: government officials and health policy advisors involved in HIV prevention strategies.Program managers: individuals overseeing PrEP implementation projects at national or regional levels.Healthcare providers: clinicians and community health workers delivering PrEP services.Community representatives: members from populations at high risk of HIV, such as adolescent girls and young women, men who have sex with men, sex workers, and individuals currently using or with lived experience using PrEP.

These stakeholders will be invited through formal invitations, ensuring a balance of representation across sectors and communities.

#### 2.6.2. Consultation Methods

To facilitate meaningful engagement, a combination of individual interviews and focus group discussions will be employed. These methods allow for in-depth exploration of stakeholder experiences and insights. Discussions will be guided by a semi-structured interview guide, developed to elicit feedback on:Preliminary findings from the scoping review.Perceived barriers and facilitators to PrEP uptake and implementation.Recommendations for enhancing PrEP programs in African contexts.

#### 2.6.3. Data Analysis and Integration

Data collected from consultations will be transcribed verbatim and analysed thematically. This analysis will identify recurring themes and insights that will be integrated into the final review. Stakeholder feedback will be used to refine interpretations and ensure that the review findings accurately reflect the realities of PrEP implementation.

#### 2.6.4. Ethical Considerations for Stage 6

Ethical approval will be obtained from a registered Research Ethics Committee (REC) in line with the National Health Act No. 61 of 2003 and NHREC guidelines. Informed consent will be obtained after providing participants with clear information about the study. Confidentiality will be ensured through anonymized data and secure storage. The study will adhere to the Declaration of Helsinki principles of respect, beneficence, and justice.

#### 2.6.5. Reporting and Dissemination

The outcomes of the consultation will be transparently reported, detailing the methods of engagement, key findings, and how stakeholder input influenced the review’s conclusions. This approach aims to enhance the credibility and applicability of the scoping review, ensuring that it serves as a valuable resource for policymakers, program implementers, and communities involved in HIV prevention efforts.

## 3. Discussion

This scoping review protocol aims to systematically map the extent, range, and nature of research concerning PrEP uptake, implementation, and associated barriers and facilitators within African contexts. By employing the Population, Concept, and Context (PCC) framework, the review seeks to provide a comprehensive overview of the existing evidence, identifying knowledge gaps and informing future research directions.

The review will encompass studies published from 2012 onwards, following the WHO’s endorsement of PrEP. A comprehensive search strategy will be developed in consultation with a research librarian, ensuring the inclusion of both published and grey literature sources. Databases such as PubMed, Scopus, CINAHL, Web of Science, African Journals Online (AJOL), and Embase will be searched, alongside grey literature from sources including the African Index Medicus, WHO publications, and national Ministry of Health websites.

The inclusion criteria are designed to capture a broad spectrum of evidence, encompassing studies focusing on PrEP implementation, uptake, adherence, barriers, facilitators, policies, programs, and service delivery models within African settings. Both quantitative and qualitative studies, as well as systematic and scoping reviews, will be considered. Publications in English, French, Arabic, or Portuguese will be included to reflect the linguistic diversity of the continent.

The data extraction process will involve charting study characteristics, PrEP implementation details, identified barriers and facilitators, outcomes, and policy and programmatic contexts. This structured approach will facilitate a comprehensive synthesis of the evidence, allowing for the identification of patterns, trends, and gaps in the literature.

While this scoping review does not aim to assess the quality of individual studies, it will provide valuable insights into the current state of PrEP research in Africa, highlighting areas where further investigation is needed. The findings will be instrumental in informing policymakers, healthcare providers, and researchers about the complexities of PrEP implementation across diverse African contexts.

## 4. Conclusions

This scoping review protocol outlines a systematic approach to mapping the landscape of PrEP uptake and implementation in Africa, emphasizing the need to understand the multifaceted barriers and facilitators that influence its success. By synthesizing existing evidence, the review aims to provide a comprehensive overview that can inform targeted interventions, policy development, and future research endeavours. Addressing the identified gaps and challenges will be crucial in enhancing PrEP accessibility and adherence, ultimately contributing to the broader goal of reducing HIV transmission across the continent.

## Figures and Tables

**Table 1 ijerph-22-01300-t001:** An example list of search terms.

PrEP and HIV prevention: “pre-exposure prophylaxis,” “PrEP,” “HIV prevention,” “HIV prophylaxis,” “antiretroviral prophylaxis,” “oral PrEP,” “long-acting PrEP,” “tenofovir,” “emtricitabine” Geographic focus: “Africa,” “Sub-Saharan Africa,” as well as specific country names (e.g., “South Africa,” “Kenya,” “Nigeria,” “Uganda,” “Tanzania,” “Ethiopia,” “Mozambique,” “Zimbabwe”) Implementation concepts: “uptake,” “adherence,” “acceptability,” “barriers,” “facilitators,” “implementation,” “rollout,” “scale-up,” “programmatic delivery,” “policy,” “guidelines,” “cost-effectiveness,” “sustainability,” “integration” Population groups: “key populations,” “men who have sex with men (MSM)” “transgender,” “adolescent girls and young women (AGYW)” “female sex workers, (FSW)” “Sero discordant couples,” “discordant couples,” “people who inject drugs (PWID)” “adolescents,” “young adults” “gays”, “bisexual” Boolean operators (AND, OR) will be applied to combine search terms, and database-specific filters will be used where available to refine results.

**Table 2 ijerph-22-01300-t002:** Population, concept, context (PCC).

Element	Description
Population	Individuals who are eligible for or using PrEP in African settings
Concept	PrEP uptake, implementation, barriers, and facilitators
Context	African countries and settings

**Table 3 ijerph-22-01300-t003:** Data charting form.

Category	Data Elements
Study ID	Author(s), year of publication, title, journal/source
Study Characteristics	Study design, objectives, country and setting, sample size, population demographics
PrEP implementation Details	Type of PrEP intervention, delivery model, duration, adherence support strategies.
Barriers and facilitators	Identified challenges and enabling factors at individual, community, health system, and policy levels.
Outcomes	National and sub-national policies, guidelines, and integration into health systems (e.g., national PrEP guidelines and integration into HIV prevention programs).
Policy and programmatic context	Information on national or sub-national policies, guidelines, and integration into health systems.
Key findings	Key findings and recommendations (e.g., effective strategies for improving PrEP uptake and policy recommendations)

## Data Availability

This scoping review protocol has been registered with the Open Science Framework (OSF) and is publicly accessible at https://osf.io/asy6g, accessed on 30 April 2025. As the protocol does not involve the generation of new data, data sharing is not applicable at this stage. However, upon completion of the scoping review, all data extracted and analyzed during the review process will be made publicly available in the OSF repository to ensure transparency and reproducibility of the research.

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
