# Peer review of "Pre-Exposure Prophylaxis Uptake, Implementation and Barriers in Africa: A Scoping Review Protocol"

_ijerph, 2025, doi:10.3390/ijerph22081300_

Round 1
Reviewer 1 Report
Comments and Suggestions for Authors
Review report for :PrEP intake, implementation and barriers in Africa: a scoping review protocol
- What is the main question addressed by the research?
- The protocol managed to identify and expand the research question which seeks to explore PrEP intake, implementation and barriers in Africa.
- Originality and relevance of the topic
- The study tackles the issue of PrEP on a broader perspective (current practices, challenges as well as the guidelines and policies followed) in an African context. The topic will assist in highlighting unaddressed issues and the findings could build upon the existing knowledge for the improvement of PrEP uptake and implementation.
- The study chosen research design is appropriate and clearly aligned with the research questions and objectives and will provide a comprehensive picture of the PrEP in Africa
- The search strategy is comprehensive, and the data selection and extraction processes are transparent.
- -The researcher has provided a clear justification for the chosen method and intends to follow established guidelines for scoping reviews, however, I would however advice
- to include potential bias and mitigating measures
- Clarify the years of the publications to be utilised- there is 2012 and 2015
- Clearly define “ other key population’ under population
- Attend to the exclusion issue as annotated on the manuscript
- The conclusion is relevant and summarises and highlights the review protocol
- All references are relevant and are in-text cited.
- General comments
This protocol followed the correct framework

Author Response
Comment 1: The researcher has provided a clear justification for the chosen method and intends to follow established guidelines for scoping reviews, however, I would however advice to include potential bias and mitigating measures.
Response 1: Thank you for your valuable feedback. We acknowledge the importance of addressing potential bias even in scoping reviews to enhance transparency and rigor. Although scoping reviews typically do not include formal risk of bias assessments like systematic reviews, we will incorporate a dedicated section outlining potential sources of bias relevant to our review, such as publication bias, language bias, and selection bias. Line 196 - 202
Comment 2: Clarify the years of the publications to be utilized- there is 2012 and 2015
Response 2: Thank you for highlighting the inconsistency regarding the timeline. We confirm that the correct start date for the inclusion of studies is 2012. The manuscript has been revised to ensure consistency throughout. Line 94, 147, 281
We selected 2012 as the starting point because this marks the year when oral PrEP was first approved by the U.S. FDA and began to be implemented in demonstration projects globally, including early efforts in some African countries. Using 2012 as the baseline allows us to capture the earliest relevant evidence on PrEP uptake, implementation, and barriers within the African context.
Comment 3: Clearly define “ other key population’ under population
Response: Thank you for your helpful observation. In response, we have removed the term “other key populations” from the Population section of the PCC framework to enhance clarity and avoid ambiguity. The revised description explicitly lists the population groups relevant to this review, based on epidemiological evidence and PrEP implementation priorities in African contexts Line 134 - 136.
Comment 4: Attend to the exclusion issue as annotated on the manuscript
Response4: Thank you for pointing out the need to clarify the exclusion criteria. We have revised the section to improve clarity and ensure alignment with the inclusion criteria. The updated Exclusion Criteria section now reads: Line 148 - 166.
Exclusion Criteria
Studies will be excluded if they:
-
Focus exclusively on the clinical efficacy or pharmacological properties of PrEP, without addressing uptake, implementation, or contextual barriers/facilitators.
-
Are conducted outside African countries or do not include African populations in multi-country studies.
-
Are opinion pieces, editorials, or commentaries that lack empirical data or analytical depth.
-
Are conference abstracts with insufficient methodological detail or results related to implementation aspects of PrEP.
Reviewer 2 Report
Comments and Suggestions for Authors
This is a needed and interesting paper/protocol and I only have a few suggestions:
Lines 39-41: Perhaps specify ORAL PrEP, as injectable PrEP has up to 100% efficacy. If you will include injectable, too, then change the word "taken" to "used."
Line 45: Remove the word "however" from the start of the sentence.
Line 123: The search terms "gay" and "bisexual" should be included in the population groups, too.
Author Response
Comment 1: Lines 39-41: Perhaps specify ORAL PrEP, as injectable PrEP has up to 100% efficacy. If you will include injectable, too, then change the word "taken" to "used."
Response 1: Thank you for your valuable feedback. The sentence has been revised to reflect that PrEP includes both oral and injectable formulations, with clarification on the consistent use and WHO endorsement. Line 39 - 43.
Comment 2: Line 45: Remove the word "however" from the start of the sentence.
Response 2: Thank you for the suggestion. The word "however" has been removed from the beginning of the sentence as recommended.
Comment 3: Line 123: The search terms "gay" and "bisexual" should be included in the population groups, too.
Response: Thank you for your comment. We have added the suggested to the list. Line 124
Reviewer 3 Report
Comments and Suggestions for Authors
This is an interesting article in which the authors have developed a protocol to help identify areas for improvement, including the implementation of strategies and the elimination of barriers in PrEP in Africa
- Title: Appropriate; it identifies the problem and the objective of the analysis.
- Abstract: Provides detailed information on its sections.
- Introduction: Well justified. The purpose and significance of the scoping review are clearly identified.
- Methods and Materials: Methodologically well structured; it provides sufficient detail to ensure reproducibility.
- Discussion: Well argued and justified regarding the need for the proposed approach.
Author Response
Response: Thank you for your positive and encouraging feedback. We appreciate your thoughtful review and are pleased to know that the manuscript met your expectations. No changes were required based on your comments.
Reviewer 4 Report
Comments and Suggestions for Authors
The paper attempts to presents a scoping review of PrEP uptake implementation and barriers across Africa.
However, this work is an incomplete submission and requires major revisions.
My comments are as shown below:-
The abstract and introduction suggest – ‘ this scoping review aims to systematically map existing literature (lines 12, 60)
and the methods suggests – ‘ The review will follow the methodological framework proposed by Arksey and O'Malley, refined by Levac et al., and will include comprehensive searches of electronic databases, grey literature, and official reports. Data will be thematically synthesized to identify research trends, gaps, and contextual variations.’ (lines 16-19, 69).
However, this work only compiles a protocol, already proposed by Arksey and O'Malley, refined by Levac et al (as the authors also state in lines 16-19, 76-78).
This work does not present any new data or results or review, and their discussion - necessary for a research or review article.
A compilation of previously described protocol does not necessitate a publication.
PRISMA provides guidance for the reporting of systematic reviews evaluating the effects of interventions. It assists authors to completely report why their systematic review was done, what methods they used, and what they found - within the context of the manuscript that includes all the data for the systematic review, and not as a separate protocol only paper - as the current manuscript is submitted.
Author Response
Comment 1: The paper attempts to presents a scoping review of PrEP uptake implementation and barriers across Africa.
However, this work is an incomplete submission and requires major revisions.
My comments are as shown below:-
The abstract and introduction suggest – ‘ this scoping review aims to systematically map existing literature (lines 12, 60)
and the methods suggests – ‘ The review will follow the methodological framework proposed by Arksey and O'Malley, refined by Levac et al., and will include comprehensive searches of electronic databases, grey literature, and official reports. Data will be thematically synthesized to identify research trends, gaps, and contextual variations.’ (lines 16-19, 69).
However, this work only compiles a protocol, already proposed by Arksey and O'Malley, refined by Levac et al (as the authors also state in lines 16-19, 76-78).
This work does not present any new data or results or review, and their discussion - necessary for a research or review article.
A compilation of previously described protocol does not necessitate a publication.
PRISMA provides guidance for the reporting of systematic reviews evaluating the effects of interventions. It assists authors to completely report why their systematic review was done, what methods they used, and what they found - within the context of the manuscript that includes all the data for the systematic review, and not as a separate protocol only paper - as the current manuscript is submitted.
Response 1: Thank you for your detailed feedback and for taking the time to review our manuscript.
We would like to respectfully clarify that the submitted manuscript is a scoping review protocol, not a completed scoping review. The intention is to outline the planned methodology for conducting the review in line with recognized guidance for protocol papers.
As stated in the title and throughout the manuscript, this is a protocol following the Arksey and O'Malley framework, further refined by Levac et al., with reporting guided by the PRISMA-ScR (Preferred Reporting Items for Systematic Reviews and Meta-Analyses extension for Scoping Reviews) checklist.
While PRISMA is indeed essential for completed systematic reviews, the PRISMA-ScR and JBI guidance support the development and publication of scoping review protocols to promote transparency, reduce duplication, and enhance methodological rigour. Publishing the protocol ensures that the review process is clearly outlined in advance and allows for peer feedback before data collection and analysis begin.
We hope this clarifies the purpose and contribution of the manuscript. We remain grateful for your time and feedback and welcome any further recommendations you may have regarding the protocol itself.
Reviewer 5 Report
Comments and Suggestions for Authors
Comments and Suggestions for Authors:
The planned review deals with the important issue of Pre-exposure prophylaxis (Prep) for the prevention of HIV-infection and Prep’s implementation among various targeted groups in African countries. Based on a thorough analysis of the available literature on this issue, the authors intend to identify research gaps and key barriers that will help to improve Prep’s implementation strategies and effective interventions across diverse African settings.
The authors give detailed justification for conducting this study. As to the Study’s Methodology the authors have well thought out the methodological framework. The strong point of the proposed study is the multilingual approach that takes into account all the information on the topic, published in different languages specific to diverse African regions.
There are a few minor flaws in the manuscript:
- Lines 76-77: it is necessary to give the authors’ names.
- Table 1, paragraph 4: it is advisable to provide explanations for abbreviations in the text, in addition to the list of abbreviations given at the end of the manuscript.
- Line 334: it seems there's a typo here.
- The abbreviations’ list is incomplete.
Author Response
Comment 1: Lines 76-77: it is necessary to give the authors’ names.
Response 1: Thank you for your comment and for pointing this out. We have revised the sentence for clarity and completeness. The updated text now reads:
“This scoping review will follow the six-stage methodological framework proposed by Arksey and O’Malley [15], and further refined by Levac et al. [16]. The review will be reported in accordance with the PRISMA-ScR (Preferred Reporting Items for Systematic Reviews and Meta-Analyses extension for Scoping Reviews) guidelines [17].” Line 77 - 80.
We believe this revision more accurately and clearly reflects the methodological guidance and reporting standards that will be followed in the conduct of the scoping review.
Comment 2: Table 1, paragraph 4: it is advisable to provide explanations for abbreviations in the text, in addition to the list of abbreviations given at the end of the manuscript.
Response 2: Thank you for your helpful suggestion. We have revised the paragraph to ensure that all abbreviations are clearly explained at first mention within the text, in addition to their inclusion in the list of abbreviations at the end of the manuscript. This change improves clarity and ensures consistency with standard reporting practices.
The updated paragraph now reads:
Population groups: “key populations,” “men who have sex with men (MSM),” “transgender,” “adolescent girls and young women (AGYW),” “female sex workers (FSW),” “serodiscordant or discordant couples,” “people who inject drugs (PWID),” “adolescents,” “young adults,” “gays,” and “bisexual individuals.” Boolean operators (AND, OR) will be applied to combine search terms, and database-specific filters will be used where available to refine results.
Comment 3: Line 334: it seems there's a typo here.
Response 3: Thank you for this comment. We have corrected the error, and it now reads AGYW -adolescent girls and young women . Line 336
Comment 4: The abbreviations’ list is incomplete.
Response 4: Thank you for your comment. We have updated the abbreviation list. Line 333 - 348
Reviewer 6 Report
Comments and Suggestions for Authors
The review protocol is well-drafted.
- The manuscript refers to two different timelines, 2012 and 2015, to include studies. Which one will be followed? Be consistent and give reasons
- The stakeholder consultation methods have not been specified - whether there will be focus group discussions, in-depth interviews, the delphi technique or any other technique/ method
- Although the stakeholder consultation includes community representatives, I suggest including PrEP users specifically.
Author Response
Comment 1: The manuscript refers to two different timelines, 2012 and 2015, to include studies. Which one will be followed? Be consistent and give reasons
Response 1: Thank you for highlighting the inconsistency regarding the timeline. We confirm that the correct start date for the inclusion of studies is 2012. The manuscript has been revised to ensure consistency throughout. We selected 2012 as the starting point because this marks the year when oral PrEP was first approved by the U.S. FDA and began to be implemented in demonstration projects globally, including early efforts in some African countries. Using 2012 as the baseline allows us to capture the earliest relevant evidence on PrEP uptake, implementation, and barriers within the African context. Line 94, 147, 280.
Comment 2: The stakeholder consultation methods have not been specified - whether there will be focus group discussions, in-depth interviews, the delphi technique or any other technique/ method.
Response 2: Thank you for your observation regarding the stakeholder consultation methods. We appreciate the opportunity to clarify this aspect of the protocol. Stakeholder consultation will be conducted through a combination of individual in-depth interviews and focus group discussions, as outlined in the “Consultation methods” section of the manuscript Line 248 -255. These qualitative approaches were selected to allow for rich, contextualized insights from a diverse group of stakeholders.
Comment 3: Although the stakeholder consultation includes community representatives, I suggest including PrEP users specifically.
Response 3: Thank you for this valuable suggestion. We agree that the inclusion of individuals with lived experience using PrEP will strengthen the stakeholder consultation by providing first-hand perspectives. While community representatives were previously included, we have now explicitly added PrEP users to this group to reflect this recommendation.
The revised section now reads:
Community representatives: members from populations at high risk of HIV, such as adolescent girls and young women, men who have sex with men, sex workers, and individuals currently using or with lived experience using PrEP. Line 242 - 245.